# Feasibility Study of CLIP-Based Key Slice Selection in CT Images and Performance Enhancement via Lesion- and Organ-Aware Fine-Tuning

**DOI:** 10.3390/bioengineering12101093

**Published:** 2025-10-10

**Authors:** Kohei Yamamoto, Tomohiro Kikuchi

**Affiliations:** 1Department of Radiology, School of Medicine, Jichi Medical University, 3311-1 Yakushi-ji, Shimotsuke, Tochigi 329-0498, Japan; 2Data Science Center, Jichi Medical University, 3311-1 Yakushi-ji, Shimotsuke, Tochigi 329-0498, Japan; r1419kt@jichi.ac.jp

**Keywords:** CLIP, CT, key slice selection

## Abstract

Large-scale medical visual question answering (MedVQA) datasets are critical for training and deploying vision–language models (VLMs) in radiology. Ideally, such datasets should be automatically constructed from routine radiology reports and their corresponding images. However, no existing method directly links free-text findings to the most relevant 2D slices in volumetric computed tomography (CT) scans. To address this gap, a contrastive language–image pre-training (CLIP)-based key slice selection framework is proposed, which matches each sentence to its most informative CT slice via text–image similarity. This experiment demonstrates that models pre-trained in the medical domain already achieve competitive slice retrieval accuracy and that fine-tuning them on a small dual-supervised dataset that imparts both lesion- and organ-level awareness yields further gains. In particular, the best-performing model (fine-tuned BiomedCLIP) achieved a Top-1 accuracy of 51.7% for lesion-aware slice retrieval, representing a 20-point improvement over baseline CLIP, and was accepted by radiologists in 56.3% of cases. By automating the report-to-slice alignment, the proposed method facilitates scalable, clinically realistic construction of MedVQA resources.

## 1. Introduction

Recent advances in vision–language models (VLMs) have been driven by joint learning from paired image–text inputs [1,2,3,4]. Medical visual question answering (MedVQA) has become a benchmark task in this domain, requiring a model to answer questions by jointly processing a medical image and the corresponding sentence(s) [5,6]. To scale training, most studies mine figures and captions from large bibliographic databases and then employ large language models (LLMs) to synthesize visual question answering (VQA) pairs [7,8]. Although effective, these bibliography-based datasets overrepresent prototypical cases and often fail to capture the heterogeneity of real-world clinical presentations, potentially reducing the model performance in practice [7,9]. Consequently, constructing MedVQA datasets directly from routine clinical data is a critical next step and could even surpass the scale of bibliography-based collections. Although a VQA dataset has been curated from routine chest radiograph reports, no comparable resource currently exists for volumetric modalities such as computed tomography (CT) or magnetic resonance imaging (MRI) [10]. A major barrier is that clinical reports are not annotated to show which sentence corresponds to which slice, and automated tools for establishing these links remain underdeveloped, hindering the construction of large, clinically sourced MedVQA datasets. Some studies have explored VLMs that use 3D encoders that pair a full image volume with the entire report; however, these models require substantial computational resources, and many state-of-the-art (SOTA) VLMs still operate on 2D inputs [11,12,13]. Moreover, everyday radiology communication—whether in report snapshots, electronic medical record attachments, or conference presentations—continues to rely on a limited set of representative 2D slices; thus, slice-level tasks remain indispensable in clinical practice.

An automated key slice selector that links each report sentence to the 2D slice that best matches its described finding is therefore essential for building the datasets needed to advance clinically useful VLMs in radiology. The key motivation of this work is to establish such a method, bridging report sentences and CT slices to enable scalable construction of clinically realistic MedVQA datasets at this scale.

This study aims to demonstrate the utility of a slice selection framework that aligns report sentences with CT slices and to investigate whether fine-tuning with lesion- and organ-aware supervision can further enhance performance. Contrastive language–image pre-training (CLIP) jointly embeds images and text into a shared latent space and computes cosine similarity between these embeddings to enable zero-shot classification and text–image retrieval [14]. We hypothesized that the same framework could be adapted to align individual report sentences with their corresponding CT slices. Each finding sentence was encoded by the text encoder, and every axial slice in the CT volume was encoded by the image encoder; the slice with the highest cosine similarity was selected as the key image for that sentence (Figure 1). This pipeline was evaluated using the original CLIP (ViT-B/16) and two medical-domain variants, PubMedCLIP and BiomedCLIP [14,15,16]. In addition, we curated a compact training set from a limited number of clinical CT studies and fine-tuned each model using this dataset.

Our contributions are summarized as follows:CLIP-based slice selector for clinical CT.We present the first CLIP-based key slice selector that can be used to construct large-scale image–text pairs directly from routine CT studies. Feasibility is evaluated with both automatic metrics and radiologist review.Efficient dataset curation protocol.We describe and release a lightweight procedure that uses a small number of studies to generate balanced fine-tuning data containing both lesion-level and organ-level (negative) sentence–slice pairs.Radiologist acceptance gains.For abnormal findings, the expert “acceptance ratio” (i.e., the proportion of automatically selected slices judged correct) is summarized in Table 1.

## 2. Related Works

### 2.1. MedVQA Dataset Construction

VQA-RAD is a foundational MedVQA benchmark for radiological imaging. Expert radiologists manually authored clinical questions and provided the corresponding ground-truth answers, ensuring high annotation quality but resulting in a relatively small dataset [5]. SLAKE builds on this by adding spatial annotations, such as pixel-level masks and bounding boxes, and by including knowledge-base-driven questions to probe deeper clinical reasoning [6]. The effective training of VLMs in highly specialized medical domains requires much larger and more diverse datasets. PMC-VQA constructs a 227 K example VQA dataset, predominantly composed of radiological images, using literature extracted from PubMed Central [7]. This dataset was built by mining images and their associated captions from articles and employing ChatGPT to generate the corresponding VQA pairs. LLava-Med adopts a similar strategy: it first trains 600K image–caption pairs mined from PubMed Central articles and then fine-tunes the model with 60K GPT-4-generated instruction-tuning examples [8]. However, none of these datasets provide explicit links between report sentences and specific CT slices, which is the key challenge that this study addresses.

### 2.2. Slice Selection

Vote-MI was proposed as a weakly supervised method for identifying representative key slices from CT volumes [17]. Although effective in highlighting diagnostically relevant slices, Vote-MI is designed for slice selection alone and does not provide a mechanism for sentence-level text–image retrieval. This limits its applicability to tasks such as MedVQA dataset construction, where fine-grained alignment between report sentences and CT slices is required. In contrast, our method explicitly addresses this gap by enabling text-driven slice retrieval and further improving alignment performance through lesion- and organ-aware fine-tuning.

Although no prior studies have applied CLIP to slice selection, related methods for extracting keyframes from sequential medical data have been employed for zero-shot surgical phase recognition [18].

### 2.3. Multimodal Learning

With the advent of CLIP, contrastive learning has become the predominant paradigm in multimodal representation learning [14]. CLIP projects textual and visual inputs into a shared embedding space by encoding each modality separately—text via a transformer-based text encoder and images via a vision transformer—and then aligns them through cosine similarity. This cross-modal alignment capability has led many researchers to employ CLIP image encoders as the visual backbone in VLMs. Subsequent studies extended this foundation to better support downstream tasks. For example, GLIP and RegionCLIP introduce region-level contrastive objectives that improve object detection and semantic segmentation performance [19,20]. SigLIP replaces the standard contrastive loss with a sigmoid cross-entropy formulation, substantially reducing the text encoder context length while maintaining high-efficiency multimodal learning [21]. CLIP-Adapter enhances few-shot transfer by inserting lightweight adapter modules into a frozen CLIP backbone, thereby boosting performance on downstream tasks without extensive retraining [22]. In the medical domain, PubMedCLIP fine-tunes CLIP using ROCO, a radiology-focused corpus derived from PubMed articles [15,23]. BiomedCLIP expands this approach by assembling a PMC-15M dataset from PubMed Central publications and applying it to further fine-tune CLIP [16]. These biomedical variants have achieved SOTA performance in cross-modal retrieval, zero-shot image classification, and MedVQA tasks, thereby establishing themselves as foundation models for a broad range of downstream medical AI applications [24,25,26].

In summary, while previous research has advanced MedVQA dataset construction, slice retrieval, and multimodal learning, none has proposed an automated framework that aligns report sentences with CT slices and enhances performance through lesion- and organ-aware fine-tuning. This constitutes the novelty of the present study.

## 3. Materials and Methods

### 3.1. Dataset

This study was approved by the Jichi Medical University Hospital Bioethics Committee for Clinical Research. We assembled our dataset from 137 consecutive patients with gastrointestinal cancer who underwent their first CT examination at our institution between 2021 and 2023. A board-certified radiologist (Radiologist 1) with more than ten years of experience reviewed each report and selected the appropriate CT series (e.g., optimal contrast phase) for each sentence describing an abnormality. The radiologist then identified the CT slice that best matched the sentence and annotated the lesion with a bounding box. The dataset was divided into 92 training, 23 validation, and 22 test studies (Figure 2). For the test set, the radiologist annotated the range of slices corresponding to each sentence to enable more detailed evaluation. This yielded 625 sentence–slice pairs for training, 152 for validation, and 120 for testing. The original reports were written in Japanese; after confirming that they contained no protected health information, all sentences were translated into English using gpt-4o-mini-2024-07-18 [27].

To improve performance on negative findings, we augmented the dataset with synthetic finding–image pairs containing no abnormalities. We first applied the TotalSegmentator to each CT series to extract organ masks and determine which of the 15 major thoracoabdominal organs were present in each slice [28]. Using these organ labels, we generated pseudo-findings in two ways: (1) inserting organ names into a fixed template and (2) prompting gpt-4o-mini to produce natural language descriptions based on the same templates. Further details of the rule- and LLM-based prompt designs are provided in Appendix A.

For lesion-positive examples, we expanded the slice range based on lesion size. Bounding boxes larger than 2000 pixels were padded by one slice above and below, whereas boxes exceeding 4000 and 6000 pixels were expanded by two and three slices, respectively. This procedure yielded a training set composed of 12,743 CT slices, including 1009 lesion-positive image pairs and 140,761 normal anatomy pairs generated from organ labels. As part of image preprocessing, we applied soft-tissue windowing to the CT values to enhance contrast in the relevant intensity ranges. Because the vision transformer (ViT) backbone requires 224 × 224 inputs, we first cropped 32 pixels from each edge of the original 512 × 512 slices to obtain 448 × 448 images and then downsampled them to 224 × 224 pixels. During training, we applied data augmentation, such as horizontal and vertical flips, translations, scaling, rotations, elastic distortions, and cutouts, to improve model robustness [29].

### 3.2. Slice Selection Algorithm

To implement this study’s approach, we defined a slice selection algorithm that links each report sentence to the most relevant CT slice. As illustrated in Figure 1, the text encoder embeds the finding sentence, while the image encoder embeds each axial slice of the CT volume. Cosine similarities are then computed between the sentence embedding and all slice embeddings. The slice with the highest similarity score is designated as the key slice corresponding to the finding. This procedure, summarized in Algorithm 1, constitutes the inference pipeline of this research method and serves as the foundation for both evaluation and fine-tuning.
**Algorithm 1** CLIP-based key slice selection.**Require:** finding sentence *f*, CT volume $V=\{s_{1},\ldots ,s_{N}\}$, text encoder $E_{\mathrm{text}}$, image encoder $E_{\mathrm{img}}$**Ensure:** key slice $s_{k}$   1:$e_{\mathrm{text}}\leftarrow E_{\mathrm{text}}\left(f\right)$   2:**for** i←1 to *N* **do**   3:    $e_{\mathrm{img}}\leftarrow E_{\mathrm{img}}\left(s_{i}\right)$   4:    ${\mathrm{sim}}_{i}\leftarrow \cos\left(e_{\mathrm{text}},e_{\mathrm{img}}\right)$   5:**end for**   6:$k\leftarrow \arg\max_{i}{\mathrm{sim}}_{i}$   7:**return** $s_{k}$

### 3.3. Models and Training Details

We trained CLIP to learn the correspondence between report sentences and CT slices. For the image encoder, we used ViT-B/16 and fine-tuned the pre-trained model with the official OpenAI weights. The same procedure was applied to PubMedCLIP and BiomedCLIP. We summarize the image encoders and text encoders used in each method in Table 2. For all text encoders, the context length was fixed at 77 tokens, and sentences exceeding this length were truncated. All training was conducted on a single NVIDIA RTX 6000 Ada GPU with a batch size of 64 using the AdamW optimizer [30]. The learning rate was scheduled from $5\times 10^{-5}$ to $1\times 10^{-6}$ over 20 epochs using cosine annealing. The checkpoint with the lowest validation loss was selected for evaluation. The experiments were run on Ubuntu 22.04.5 LTS with Python 3.11.9 (conda-forge), using numpy 2.1.1, torch 2.4.1+cu118, and TotalSegmentator 2.10.0.

## 4. Results

We evaluated the slice selection performance of six models: pre-trained CLIP, PubMedCLIP, and BiomedCLIP, along with their fine-tuned counterparts (FT).

Before reporting task-specific retrieval metrics, we first verified whether each model learned meaningful alignments during training. To this end, we measured the mean absolute error (MAE) between the predicted and ground-truth slice indices using two strategies: hard prediction (selecting the Top-1 slice) and soft prediction (using a local moving average).

In Section 4.2, we assess the retrieval accuracy of sentences describing abnormal findings from diagnostic reports. In addition, we report on a radiologist-led evaluation of slice selection results to examine clinical applicability. Section 4.3 presents the alignment fidelity between organ-related text and its corresponding CT slices. Finally, Section 4.4 provides visualizations of the slice selection results for key series, highlighting both quantitative performance and qualitative interpretability.

### 4.1. Training Verification via Slice-Level MAE

To validate the training process, we measured the mean absolute error (MAE) between the predicted and ground-truth slice indices across all models using the image–sentence pairs from the validation set. We compared two strategies: (i) hard prediction, which directly selects the slice with the highest similarity score, and (ii) soft prediction, which applies a moving average over ±2 neighboring slices to leverage the sequential continuity of CT volumes. As shown in Figure 3, fine-tuned models consistently achieved a lower MAE than their pre-trained counterparts, confirming that meaningful learning occurred. Moreover, the soft prediction strategy yielded fewer errors than the hard prediction strategy, motivating its adoption for all subsequent evaluations. For example, in BiomedCLIP, the MAE improved from 10.40±13.58 with hard prediction to 9.21±11.09 with soft prediction, corresponding to an average gain of approximately one slice in localization accuracy.

### 4.2. Lesion Awareness

The models are evaluated using a test set of 22 studies containing 120 sentences with manually annotated ground-truth slice ranges. At inference, each sentence is encoded by the CLIP text encoder, every slice in the corresponding CT volume is processed by the image encoder, cosine similarities are computed, and the slice with the highest score is selected as the model’s key slice (Figure 1, Algorithm 1). To assess interobserver variability, Radiologist 2—independent of the ground-truth annotator—also selected the single best-matching slice for each sentence. Table 3 shows the proportion of Top-1 predictions falling within the annotated ranges and the proportion of cases in which any of the Top-5 most similar slices overlapped with those ranges. CLIP (FT) improves the Top-1 accuracy by 20 percentage points over CLIP, demonstrating effective adaptation to the medical domain. PubMedCLIP and BiomedCLIP began with higher baseline accuracies, which are further improved by fine-tuning. BiomedCLIP (FT), in particular, achieves the best performance, with a Top-1 accuracy of 51.72% and Top-5 accuracy of 64.37%. Considering that Radiologist 2 achieved a Top-1 accuracy of 78.16%, the model accuracy of 51.72% represents a relatively strong result.

As noted above, strict ground-truth ranges do not always align with radiologists’ subjective assessments. To evaluate whether the slices proposed by each method would be considered acceptable in clinical practice, we conducted a qualitative assessment in which a radiologist evaluated whether to adopt each suggested slice for a given sentence. We developed a simple web application that displayed each finding sentence alongside the slices proposed by each slice selector. For each case, the suggestions from all models were presented in a random order on the same screen, and the radiologist was blinded to the source model. Table 4 reports the adoption rates for each model’s recommendations. Although these adoption rates generally correspond to the quantitative results presented in Table 3 (Top-1 accuracy), they are uniformly slightly higher. These higher adoption rates occur when lesions deemed acceptable by the radiologist fall outside the strict ground-truth range—for example, calcifications extending across several slices or annotation covering only one of multiple hepatic cysts.

### 4.3. Organ Awareness

Recognizing normal anatomy is as important as detecting abnormalities, so we define organ-aware slice selection as a complementary evaluation. Using TotalSegmentator, we generated binary organ presence labels for each CT slice based on the 15 major thoracoabdominal organs [28]. For each model, we encoded the organ prompt and each CT slice, computed their cosine similarities, and then determined organ-specific thresholds using the Youden index on the validation set, which served as cutoffs for determining organ presence in each test slice [31]. To assess the impact of text granularity, we compared two input types: single-word organ names (e.g., “Heart” and “Liver”) and full template sentences (e.g., “This CT image includes the heart”). These experiments test whether word- or sentence-level prompts are more effective when querying a text encoder.

Table 5 summarizes the organ-aware evaluation, reporting both the accuracy and F1-score for each model under word and sentence prompt conditions. BiomedCLIP (FT) achieved the highest performance in both settings, with an F1-score of 0.85 for word and sentence inputs. PubMedCLIP showed similar accuracy and F1-scores across the two prompt types, while BiomedCLIP tended to perform better with sentence prompts. Fine-tuning yielded comparable gains for word and sentence inputs across all models, indicating that our approach generalized well to both single-token organ names and fully descriptive sentences. On average, fine-tuning improved the accuracy and F1-score by more than 10% for each model. These substantial improvements demonstrate enhanced normal anatomy recognition accuracy not only for the original CLIP model but also for the medical-domain pre-trained variants (PubMedCLIP and BiomedCLIP). They further suggest that fine-tuning improves slice extraction accuracy for non-abnormal findings, thereby benefiting downstream VQA generation. F1-scores for each organ are provided in Appendix B.

Additionally, as an external dataset benchmark, we evaluated organ awareness using the publicly available CT-RATE dataset, which consists of chest CT volumes with organ labels generated by TotalSegmentator [13]. We evaluated 1304 cases from the validation split, selecting the earliest series (denoted as “_a_1” in the dataset) when multiple CT series were available per patient. The results are summarized in Table 6. Overall performance was lower than that reported in Table 5, likely due to domain shifts related to imaging devices and acquisition parameters. For example, BiomedCLIP(FT) achieved an accuracy of 0.926 and an F1-score of 0.809 on CT-RATE, compared with 0.948 and 0.854, respectively, on the in-house dataset. Despite this decrease, BiomedCLIP(FT) consistently yielded the highest accuracy across all models, confirming its robustness to dataset variation.

### 4.4. Visualization

Figure 4 shows an example of a lesion-aware slice selection result on the test cohort. For uterine fibroids, PubMedCLIP, BiomedCLIP, and BiomedCLIP(FT) correctly identified key CT slices containing the lesion. Figure 5 shows similarity profiles between organ-related text and CT slices for a selected series. The original CLIP model exhibited numerous false positives and inconsistent alignments, whereas CLIP (FT) produced a much cleaner correspondence and improved localization. Although PubMedCLIP and BiomedCLIP also showed false positives and negatives, both models achieved high accuracy in recognizing the target organ after fine-tuning. Additional qualitative examples are presented in Appendix C.

## 5. Discussion

In this study, we introduced a CLIP-based key slice selector that aligns free-text radiology findings with corresponding CT slices. The approach was evaluated along two complementary axes: lesion and organ awareness. Despite training on a relatively small dataset, the fine-tuned models achieved substantial gains in lesion-aware slice retrieval, improving accuracy by 7–20 percentage points over the pre-trained baselines. The best performer, BiomedCLIP (FT), achieved 51% Top-1 accuracy for lesion localization. In the organ-aware task, all fine-tuned models achieved improvements of over 10% in both accuracy and F1-score. Notably, the performance remained strong regardless of whether organ prompts were provided as single words or full sentences, demonstrating the flexibility of the text-encoding strategy.

To further examine the robustness under external conditions, we conducted an additional organ-awareness evaluation on the publicly available CT-RATE dataset. Although all methods exhibited some degradation in performance owing to domain shifts in imaging devices and acquisition protocols, the decrease was relatively modest, and BiomedCLIP (FT) consistently achieved the highest accuracy. These findings suggest that this study’s approach is robust across datasets and can be generalized beyond the in-house cohort.

Regarding system requirements, all experiments were conducted on a single NVIDIA RTX 6000 Ada GPU with 48 GB of memory, which was sufficient to train models with a batch size of 64. Each training run with 20 epochs finished within one hour. These results indicate that the proposed method can be reproduced on a high-end single-GPU workstation without requiring distributed resources.

Beyond the dataset used in this study, the proposed slice selection framework offers practical benefits when applied to diverse imaging archives. First, automatically aligning report sentences with key slices can substantially reduce the cost of two-dimensional (2D) slice-level annotation, which is often a major bottleneck in dataset curation. Second, it enables the construction of image–text pairs directly from unlabeled CT volumes and routine reports, providing a low-cost pathway for expanding training resources for medical VLMs. These improvements highlight the broader utility of the proposed method across datasets and institutions, extending its impact beyond the present feasibility study.

The method paves the way for fully automated construction of MedVQA datasets from routine clinical archives, directly addressing the biases and scale limitations inherent in bibliographic collections. Furthermore, the radiologist’s evaluation of the slice selection results (Table 4) and visualization of similarity profiles (Figure 5) suggest that, in addition to serving as a VQA dataset generator, integrating the slice selector into clinical workflows as a recommendation tool could help reduce radiologists’ workload and enhance diagnostic accuracy.

This study has several limitations that should be acknowledged. First, as this was a feasibility study, the training and test cohorts were small and drawn from a single institution. Expanding the dataset size could uncover additional gains in retrieval accuracy and reduce the risk of overfitting to the relatively small evaluation cohorts. Second, the evaluation reports only overall lesion- and organ-aware metrics; it does not stratify the results by lesion size, anatomical region, contrast phase, or disease category. Finer-grained analysis is essential to uncover failure modes in clinically relevant subgroups. Third, the Youden index thresholds used to determine organ presence were chosen from the validation set and fixed for testing. Although this mirrors a real-world deployment scenario, alternative calibration strategies may yield better threshold stability for unseen data.

In future work, we will integrate a key slice selector into a complete MedVQA generation pipeline consisting of four main stages.

Segmenting diagnostic reports into individual finding units;Acquiring the corresponding imaging series (including selection of contrast phases);Extracting key slices using the slice selector;Generating VQA pairs from each finding sentence through rule-based systems or LLMs.

While this study addresses Stage 3, the remaining steps are critical for end-to-end automation. Accordingly, we plan to develop and evaluate methods for report segmentation and image retrieval and to implement VQA generation using both deterministic templates and generative LLM prompts. As these components mature, we will iteratively expand the clinical dataset and refine the slice selector’s accuracy.

We also plan to train VLMs on the resulting automatically constructed MedVQA dataset and compare their performance with models trained on manually curated slice annotations to quantify the impact of automated slice selection on the downstream VLM capabilities.

Beyond dataset construction, architectural advances offer promising directions. Recent large-scale VLMs such as Merlin, which align 3D CT volumes with radiology reports through contrastive learning, provide useful design principles [12]. Incorporating similar volume-level representation strategies, as well as techniques from the CLIP-Driven Universal Model—which uses text embeddings as semantic labels for segmentation and detection—could extend the framework beyond slice retrieval to fine-grained localization and more efficient MedVQA dataset construction [32].

## 6. Conclusions

This study introduced a CLIP-based key slice selector that aligns free-text radiology findings with the corresponding CT images. Fine-tuning on a compact dual-supervised dataset substantially improved lesion-aware accuracy, with the best-performing model achieving strong results in lesion localization. In the organ-aware task, all fine-tuned models demonstrated gains in both accuracy and F1-score, regardless of whether the prompts were provided as single words or full sentences. A qualitative review indicated that the slice selector’s recommendations could reduce radiologists’ workload and enhance diagnostic confidence in routine practice. Nevertheless, the current evaluation is limited by its single-institution, small-cohort design, lack of stratification by lesion size or disease type, and reliance on fixed-threshold calibration, all of which limit generalizability. Future work will expand the dataset, incorporate stratified analyses, and integrate the selector into an end-to-end MedVQA pipeline encompassing report segmentation, series retrieval, slice extraction, and automatic question–answer generation.

## Figures and Tables

**Figure 1 bioengineering-12-01093-f001:**
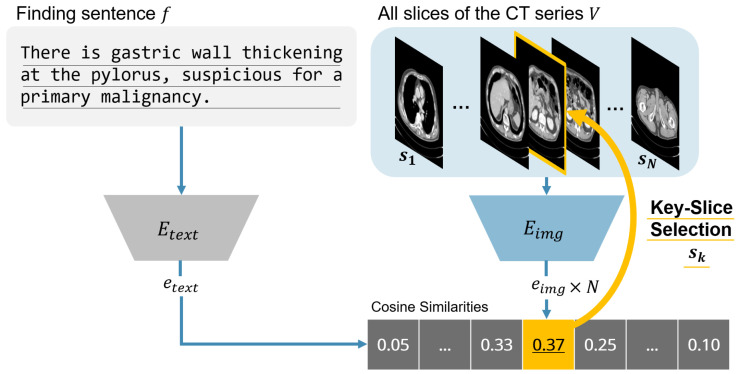
Overview of the proposed contrastive language–image pre-training (CLIP)-based slice selection pipeline. Each report sentence is encoded by the text encoder and each CT slice by the image encoder. Cosine similarities are computed between the sentence embedding and all slice embeddings, and the slice with the highest score is chosen as the key slice for that finding.

**Figure 2 bioengineering-12-01093-f002:**
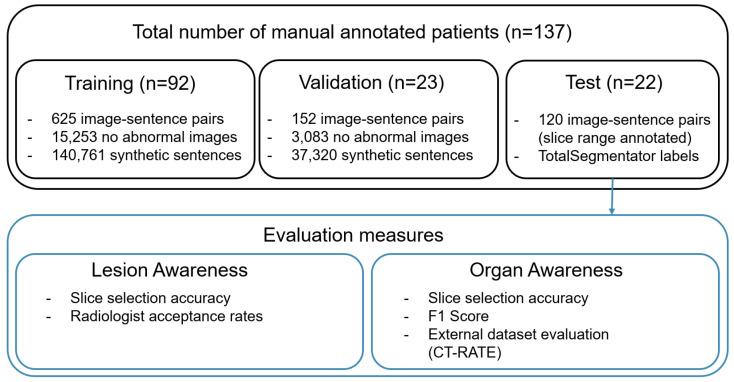
Overview of dataset composition and evaluation measures.

**Figure 3 bioengineering-12-01093-f003:**
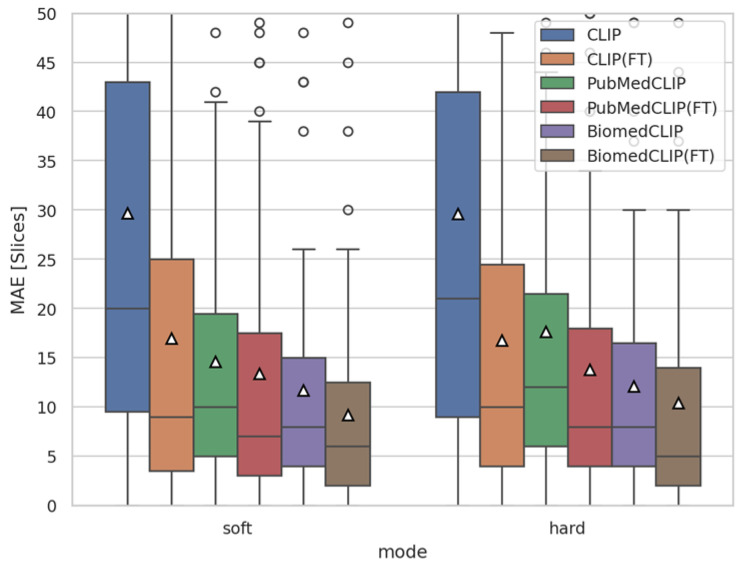
Comparison of mean absolute error (MAE) between predicted and ground-truth slice indices for each method. Results are shown for both hard prediction (Top-1 slice) and soft prediction (moving average over ±2 slices).

**Figure 4 bioengineering-12-01093-f004:**
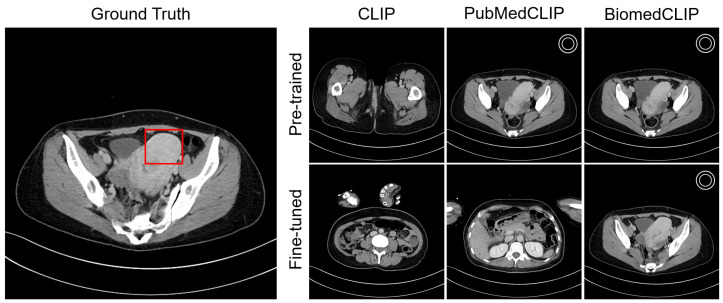
“Multiple masses that may represent fibroids are observed in the myometrium and beneath the endometrium.” The Ground Truth column shows the reference CT slice with the radiologist-annotated lesion bounding box. The other columns display each model’s key slice prediction. The white double circle symbol indicates predictions that match the ground-truth slice range.

**Figure 5 bioengineering-12-01093-f005:**
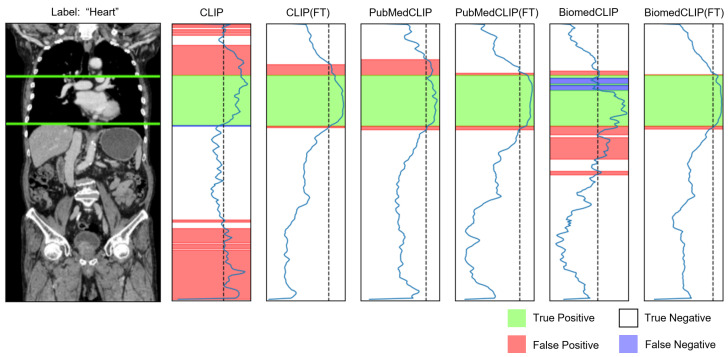
Similarity profiles between the prompt “Heart” and each slice. The green line denotes the ground-truth slice range. Similarity profiles increase from left to right, and the dashed line indicates the decision threshold.

**Table 1 bioengineering-12-01093-t001:** Radiologist acceptance ratio for automatically selected slices with abnormal findings.

Model	Baseline [%]	+ Fine-Tuning [%]	Δ [%]
CLIP	24.14	40.23	+16.09
PubMedCLIP	37.93	50.57	+12.64
BiomedCLIP	50.57	56.32	+5.75

**Table 2 bioengineering-12-01093-t002:** Summary of image and text encoders used in each method.

Method	Image Encoder	Text Encoder
CLIP	ViT-B/16	Transformer
PubMedCLIP	ViT-B/32	Transformer
BiomedCLIP	ViT-B/16	PubMedBERT

**Table 3 bioengineering-12-01093-t003:** Comparison of slice selection accuracy for each method. Reported values are Top-1 and Top-5 accuracy [%]. For each metric, the highest score among the automated methods (excluding Radiologist 2) is highlighted in red.

Method	Acc.@1 ↑	Acc.@5 ↑
Radiologist2	78.16	-
CLIP	19.54	44.83
CLIP(FT)	40.23	49.43
PubMedCLIP	29.89	54.02
PubMedCLIP(FT)	42.53	59.77
BiomedCLIP	44.83	60.92
BiomedCLIP(FT)	51.72	64.37

**Table 4 bioengineering-12-01093-t004:** Radiologist acceptance rates [%] of slice selector predictions for each model.

Method	Acceptance Rate ↑
CLIP	24.14
CLIP(FT)	40.23
PubMedCLIP	37.93
PubMedCLIP(FT)	50.57
BiomedCLIP	50.57
BiomedCLIP(FT)	56.32

**Table 5 bioengineering-12-01093-t005:** Comparison of organ extraction performance. Reported metrics are accuracy (Acc.) and F1-score (F1). “Word” indicates prompts using single organ names, and “Sentence” indicates prompts using full finding sentences. The highest score in each column is highlighted in red.

Method	Word		Sentence	
	**Acc. ↑**	**F1 ↑**	**Acc. ↑**	**F1 ↑**
CLIP	0.715±0.105	0.514±0.182	0.725±0.071	0.520±0.192
CLIP(FT)	0.927±0.038	0.806±0.153	0.934±0.035	0.824±0.129
PubMedCLIP	0.881±0.046	0.722±0.164	0.882±0.042	0.726±0.157
PubMedCLIP(FT)	0.943±0.030	0.839±0.123	0.944±0.031	0.843±0.121
BiomedCLIP	0.832±0.133	0.673±0.230	0.884±0.048	0.730±0.126
BiomedCLIP(FT)	0.947±0.033	0.853±0.118	0.948±0.032	0.854±0.118

**Table 6 bioengineering-12-01093-t006:** Comparison of organ extraction performance on the external CT-RATE dataset. Reported metrics are accuracy (Acc.) and F1-score (F1). “Word” indicates prompts using single organ names, and “Sentence” indicates prompts using full finding sentences. The highest score in each column is highlighted in red.

Method	Word		Sentence	
	**Acc. ↑**	**F1 ↑**	**Acc. ↑**	**F1 ↑**
CLIP	0.726±0.059	0.554±0.223	0.724±0.065	0.556±0.226
CLIP(FT)	0.902±0.050	0.770±0.251	0.902±0.048	0.770±0.250
PubMedCLIP	0.842±0.099	0.680±0.237	0.861±0.095	0.707±0.226
PubMedCLIP(FT)	0.911±0.072	0.796±0.252	0.910±0.073	0.795±0.252
BiomedCLIP	0.749±0.171	0.632±0.271	0.755±0.125	0.604±0.226
BiomedCLIP(FT)	0.926±0.059	0.809±0.257	0.926±0.059	0.809±0.257

## Data Availability

The datasets generated and analyzed during the current study are not publicly available due to privacy and ethical restrictions related to patient medical imaging data. The source code used for the analysis is available from the corresponding author upon reasonable request.

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
