# Peer review of "Feasibility Study of CLIP-Based Key Slice Selection in CT Images and Performance Enhancement via Lesion- and Organ-Aware Fine-Tuning"

_bioengineering, 2025, doi:10.3390/bioengineering12101093_

Round 1

Reviewer 1 Report

Comments and Suggestions for Authors

Comparision with Benchmark datasets would be expected 

More emphasize on methods is necessary 

dataset discription is necessary train and validation, how the dataset preprocessing and selection for training 

Provide results on proposed dataset and compare with other similar dataset 

discuss the improvements of proposed methods on various data and your dataset as well

Discuss the system requirements and provide environment setup for this experiment 

refrence 12 experimental outline could be adopted for further modifications

cite and discuss {

https://www.sciencedirect.com/science/article/pii/S136184152400210X

https://www.sciencedirect.com/science/article/pii/S1361841524002950

https://www.sciencedirect.com/science/article/pii/S1361841525000179

https://doi.org/10.1016/j.media.2024.103370

Liu, Jie, et al. "Clip-driven universal model for organ segmentation and tumor detection." Proceedings of the IEEE/CVF international conference on computer vision. 2023.

}

 Ablation study could be helpful for highkighting contributions

Author Response

## Reviewer 1: 
General Response: We sincerely thank the reviewer for their thorough evaluation and valuable feedback. We have carefully revised the manuscript to address these comments and believe that the quality has been substantially improved.

Comments 1: Comparision with Benchmark datasets would be expected 
Response 1: We have added an external dataset benchmark in Section 4.3 (Organ Awareness, lines 248–257) using the CT-RATE dataset. The corresponding results are presented in the newly added Table 5. This enables future studies to directly compare their methods with ours using a standardized benchmark setting.

Comments 2: More emphasize on methods is necessary 
Response 2: We have addressed this by restructuring the Methods section to place a stronger emphasis on the proposed approach. In particular, we have added a dedicated subsection (Section 3.2, Slice-Selection Algorithm, lines 148–155) that explicitly describes the algorithmic flow of our method, complemented by Figure 1 and Algorithm 1, thereby highlighting the methodological contributions more clearly.

Comments 3: dataset discription is necessary train and validation, how the dataset preprocessing and selection for training 
Response 3: In Section 3.1 (Dataset), we added Figure 2 to provide a detailed description of the dataset, including the training/validation split and the preprocessing and selection procedures for training.

Comments 4: Provide results on proposed dataset and compare with other similar dataset 
Response 4: We added a comparison with an external dataset (CT-RATE) to the Results section (lines 248–257). The results demonstrate that our method achieved performance trends comparable to those observed on the proposed in-house dataset, thereby confirming the robustness and generalizability of the approach across datasets.

Comments 5: discuss the improvements of proposed methods on various data and your dataset as well
Response 5: In the Discussion (lines 279–284), we have added a paragraph that discusses the broader improvements that our method can bring when applied to other datasets, including reducing the cost of 2D annotations and enabling low-cost construction of image–text pairs from unlabeled data.

Comments 6: Discuss the system requirements and provide environment setup for this experiment 
Response 6: In the Discussion (lines 285-289), we have added a paragraph describing the system requirements, including the GPU memory and training time. In Section 3.3 (Models), we expand the description of the experimental environment to include details of the operating system, Python, and library versions, and preprocessing tools (lines 162-167).

Comments 7: refrence 12 experimental outline could be adopted for further modifications
Response 7: In the Future Work section, we have added a discussion on Merlin (Ref. 12), a recently proposed vision language model that aligns 3D CT volumes with radiology reports using contrastive learning (lines 330–332). We noted that incorporating volume-level representation strategies similar to Merlin could further improve the accuracy of our slice-selection framework and facilitate a more efficient construction of VQA datasets.

Comments 8: cite and discuss {https://www.sciencedirect.com/science/article/pii/S136184152400210X
https://www.sciencedirect.com/science/article/pii/S1361841524002950
https://www.sciencedirect.com/science/article/pii/S1361841525000179
https://doi.org/10.1016/j.media.2024.103370
Liu, Jie, et al. "Clip-driven universal model for organ segmentation and tumor detection." Proceedings of the IEEE/CVF international conference on computer vision. 2023.}

Response 8: We have carefully reviewed all the suggested references. The first four articles were not directly related to our specific research focus, and incorporating them would make the narrative less clear to readers; therefore, we did not cite them. However, we found the fifth reference (Liu et al., CLIP-Driven Universal Model for Organ Segmentation and Tumor Detection, ICCV 2023) to be highly relevant. Therefore, we have added a discussion in the Future Work section, highlighting how techniques from this study could be incorporated into our framework to extend slice-level retrieval toward fine-grained localization tasks such as detection and segmentation(lines 332–336).

Comments 9: Ablation study could be helpful for highkighting contributions
Response 9: In Appendix D, we added an ablation study entitled Impact of Organ-Aware Synthetic Data, which evaluates lesion-aware performance with and without synthetic data.

Reviewer 2 Report

Comments and Suggestions for Authors

The authors submitted a manuscript entitled “Feasibility Study of CLIP-Based Key Slice Selection in CT Images and Performance Enhancement via Lesion- and Organ-Aware Fine-Tuning”. After a careful review, I recommend accepting the paper after major revisions, subject to the incorporation of the following comments:

  1. The manuscript requires further refinement in grammar and adherence to academic writing tone.
  2. While the authors mention the novelty, the abstract should be strengthened by including details on performance metrics such as accuracy and overall model effectiveness.
  3. The research motivation should be discussed more clearly, and specific research questions should be explicitly highlighted.
  4. The related work section lacks sufficient discussion and analysis. Although an adequate number of references are cited, the novelty compared to prior studies must be emphasized and critically analyzed.
  5. Regarding the Ethical Approval (Section 3.1), the statement “This study was approved by the ethics review board of our institution” is vague. Please provide the name of the institution that granted IRB approval. Additionally, avoid the use of first-person pronouns such as “we,” “my,” “I,” and “our” throughout the manuscript.
  6. Regarding the Model and Training Details (Section 3.2), this section requires more thorough explanation. A clear protocol for training and testing should be provided. The proposed methodology and model architecture should be presented in a graphical format for clarity. Further details are needed on model development, optimization, training, testing, and validation. The discussion of CLIP alone is insufficient.
  7. Algorithm 1 must be included in the methodology section, discussed in detail, and cited appropriately in the main text.
  8. To support each step of the methodology, every major point should be verified and validated in the results section, using appropriate graphs, plots, charts, or other visual evidence.

Author Response

## Reviewer 2:

The authors submitted a manuscript entitled “Feasibility Study of CLIP-Based Key Slice Selection in CT Images and Performance Enhancement via Lesion- and Organ-Aware Fine-Tuning”. After a careful review, I recommend accepting the paper after major revisions, subject to the incorporation of the following comments:
General Response: We appreciate the reviewer’s constructive suggestions and the detailed review. In the revised version, we have made extensive revisions to improve clarity, grammar, and overall presentation as recommended.

Comment 1: The manuscript requires further refinement in grammar and adherence to academic writing tone.
Response 1: We have used a proofreading service to ensure that the manuscript has been carefully reviewed.

Comments 2: While the authors mention the novelty, the abstract should be strengthened by including details on performance metrics such as accuracy and overall model effectiveness.
Response 2: We have revised the Abstract to include specific performance metrics (e.g., Top-1 accuracy of 51.7% and radiologist acceptance rate of 56.3%) to better highlight the effectiveness of our approach (lines 12-14).

Comments 3: The research motivation should be discussed more clearly, and specific research questions should be explicitly highlighted.
Response 3: We have revised the Introduction to explicitly clarify the research motivation in the first paragraph (lines 41–45), emphasizing the need for an automated method to bridge report sentences and CT slices for clinically realistic MedVQA dataset construction. Furthermore, in the second paragraph (lines 46–48), we explicitly state the research question by outlining the focus of this study as the utility of a slice-selection framework and the effects of lesion- and organ-aware fine-tuning.

Comments 4: The related work section lacks sufficient discussion and analysis. Although an adequate number of references are cited, the novelty compared to prior studies must be emphasized and critically analyzed.
Response 4: We have revised the Related Work section to provide a more critical analysis. In particular, we highlighted the limitation of Vote-MI, which focuses solely on key-slice selection without supporting sentence-level text–image retrieval, and clarified that our method explicitly addresses this gap. In addition, we have added a concluding paragraph summarizing the gaps between prior studies and emphasizing the novelty of our approach (lines 109-112).

Comments 5: Regarding the Ethical Approval (Section 3.1), the statement “This study was approved by the ethics review board of our institution” is vague. Please provide the name of the institution that granted IRB approval. Additionally, avoid the use of first-person pronouns such as “we,” “my,” “I,” and “our” throughout the manuscript.
Response 5: We have revised the section related to Ethics Approval. Throughout the manuscript, occurrences of first-person pronouns have been reviewed and either removed or retained only where their use is particularly necessary, with professional proofreading ensuring consistency and formal tone.

Comments 6: Regarding the Model and Training Details (Section 3.2), this section requires more thorough explanation. A clear protocol for training and testing should be provided. The proposed methodology and model architecture should be presented in a graphical format for clarity. Further details are needed on model development, optimization, training, testing, and validation. The discussion of CLIP alone is insufficient.
Response 6: We addressed this comment in two ways. First, we used Algorithm 1 to describe the training and inference protocols in detail and explicitly clarified that the same process applies to both stages. Second, although we did not propose a new model architecture, making a graphical illustration redundant, we agree that the architecture of the models employed should be stated more clearly. Therefore, in Section 3.3 (Models), we added a table summarizing the image and text encoders used in each method (Table 1, lines 148-155).

Comments 7: Algorithm 1 must be included in the methodology section, discussed in detail, and cited appropriately in the main text.
Response 7: Algorithm 1 has been moved from the Results section to the Methods section. We also expanded the accompanying text to provide a detailed explanation of the inference procedure and ensure that Algorithm 1 is properly cited and discussed in the main text.

Comments 8: To support each step of the methodology, every major point should be verified and validated in the results section, using appropriate graphs, plots, charts, or other visual evidence.
Response 8: To address this comment, we have added a new subsection to the Results section, presenting a boxplot of the mean absolute error (MAE) between the predicted and ground-truth slice indices on the validation set (lines 182-193). We compared the hard predictions (top-1 slice) and soft predictions (moving average over neighboring slices) across all models. The results (Fig. 3) show that fine-tuning consistently reduced the MAE and that the soft prediction strategy yielded lower errors than hard prediction, thereby verifying that the training process led to meaningful learning. This step supported the validity of our methodology before proceeding with task-specific evaluations.

Round 2

Reviewer 1 Report

Comments and Suggestions for Authors

Do update the manuscript with highlights and major contributions of proposed work. 

Addition of graphical abstract is appriciated.

Rest all queries were addressed 

Author Response

## Reviewer 1:

General Response: We have carefully addressed the two comments provided by the reviewer, and we would greatly appreciate your confirmation.

Comments 1: Do update the manuscript with highlights and major contributions of proposed work.

Response 1: We have added a paragraph summarizing the major contributions of this work in the Introduction section (lines 59-70).

Comments 2: Addition of graphical abstract is appriciated.

Response 2: We have created a graphical abstract that concisely summarizes our study and submitted it under the "Graphical Abstract" section in the submission form.

Reviewer 2 Report

Comments and Suggestions for Authors

The authors addressed all the comments. 

Author Response

## Reviewer 2:

General Response: We are pleased to have addressed all comments from Reviewer 2 in Round 1. In Round 2, we have responded to the remaining comments from Reviewer 1. We sincerely appreciate your thorough review and valuable feedback.